# Sweet Immunity: Inulin Boosts Resistance of Lettuce (*Lactuca sativa*) against Grey Mold (*Botrytis cinerea*) in an Ethylene-Dependent Manner

**DOI:** 10.3390/ijms20051052

**Published:** 2019-02-28

**Authors:** Łukasz Paweł Tarkowski, Bram Van de Poel, Monica Höfte, Wim Van den Ende

**Affiliations:** 1Laboratory of Molecular Plant Biology, KU Leuven, Kasteelpark Arenberg 31, 3001 Leuven, Belgium; lukaszpawel.tarkowski@kuleuven.be; 2Laboratory of Molecular Plant Hormone Physiology, Division of Crop Biotechnics, Department of Biosystems, KU Leuven, 3001 Leuven, Belgium; bram.vandepoel@kuleuven.be; 3Laboratory of Phytopathology, Department of Plants and Crops, UGhent, 9000 Ghent, Belgium; monica.hofte@ugent.be

**Keywords:** sweet immunity, *Botrytis cinerea*, fructan, oligogalacturonides, lettuce, ethylene, 1-MCP, GABA

## Abstract

The concept of “Sweet Immunity” postulates that sugar metabolism and signaling influence plant immune networks. In this study, we tested the potential of commercially available inulin-type fructans to limit disease symptoms caused by *Botrytis cinerea* in lettuce. Spraying mature lettuce leaves, with inulin-type fructans derived from burdock or chicory was as effective in reducing grey mold disease symptoms caused by *Botrytis cinerea* as spraying with oligogalacturonides (OGs). OGs are well-known defense elicitors in several plant species. Spraying with inulin and OGs induced accumulation of hydrogen peroxide and levels further increased upon pathogen infection. Inulin and OGs were no longer able to limit *Botrytis* infection when plants were treated with the ethylene signaling inhibitor 1-methylcyclopropene (1-MCP), indicating that a functional ethylene signaling pathway is needed for the enhanced defense response. Soluble sugars accumulated in leaves primed with OGs, while 1-MCP treatment had an overall negative effect on the sucrose pool. Accumulation of γ-aminobutyric acid (GABA), a stress-associated non-proteinogenic amino acid and possible signaling compound, was observed in inulin-treated samples after infection and negatively affected by the 1-MCP treatment. We have demonstrated for the first time that commercially available inulin-type fructans and OGs can improve the defensive capacity of lettuce, an economically important species. We discuss our results in the context of a possible recognition of fructans as Damage or Microbe Associated Molecular Patterns.

## 1. Introduction

In a rapidly changing world, there is a need for high yielding, stress tolerant crops. Pest attacks account for an estimated loss of $20–40 billion worldwide [1]. Besides classical breeding efforts, the use of chemical pesticides is widespread, although toxicity risks for human health and the environment are an issue [2], and production costs are growing exponentially [3]. Sustainable alternatives could lie in “biological control” strategies [4] such as the use of beneficial microorganisms [5] or ‘priming’ plants by spraying natural compounds [6]. Priming helps the plant to react faster and/or more strongly to a future pathogen attack, and is believed to work with minimal energy inputs, triggering crop resistance to (a)biotic stresses without major implications for plant fitness [7]. Priming can be useful in the field and greenhouse context on intact plants, but also for postharvest application on fruits [8] and detached leaves [9]. In this context, it is important to note that inulin is a safe product for human consumption, also possessing prebiotic properties [10]. 

In primed plants, plant immune signaling pathways are activated after pathogen infection, involving all the major defense phytohormones: jasmonic acid (JA), salicylic acid (SA), abscisic acid (ABA), and ethylene in a pathosystem-specific manner [11,12,13]. In general, it is accepted that ethylene, JA, and ABA play a major role in modulating defense responses against necrotrophs [14,15], whereas SA is known to be critical for defense against biotrophs [16,17]. However, exceptions to this dichotomy are observed [18,19]. The role of ethylene may also depend on the physiological status of the plant, the plant species and the exact status of the sampled materials [20,21]. The chemical nature of the priming compounds characterized so far is very heterogeneous, including amino acids, organic acids, inorganic molecules, and other categories [22,23,24]. Among those, carbohydrates recently gained considerable attention [25,26] because of related advantages such as low production costs and absence of toxicity. Carbohydrates are the products of photosynthesis, and besides their fundamental metabolic role, the signaling properties of molecules such as hexoses and sucrose (Suc) are well-established [27,28]. Soluble sugar signaling and dynamics are crucial for the control of plant development and organogenesis [29,30,31], as well as for coping with biotic and abiotic stresses [32,33,34]. Their emerging importance in immune processes lead to the development of the “Sweet Immunity” concept, which postulates that sugars and components of the sugar metabolism are essential players in plant defense strategies [35]. Attempts to use soluble carbohydrates such as cell wall-derived oligogalacturonides (OGs) [36] and cellobiose [37] as priming agents led to important successes. However, few research efforts focused on priming or treatments with non-structural soluble sugars [25,38].

Soluble sugar levels are tightly controlled by transporters and enzymes. Among them, invertases are able to irreversibly split Suc into glucose (Glc) and fructose (Fru) and play a central role in source-sink partitioning and plant development [39,40]. These enzymes are important players in the plant adaptation strategies towards biotic and abiotic stresses [41,42,43], and can be classified in acidic and alkaline forms [44].

Here, we focus on fructans, which accumulate in approximately 15% of land plants. These oligo- and polysaccharides are synthesized by elongating the Fru part of Suc with Fru moieties via β(2-1) and/or β(2-6) linkages, resulting in three major subclasses: inulins (β(2-1) bonds), levans (β(2-6) bonds), and graminans (mixed bonds) [45]. Fructans can also be synthesized by adding a Fru moiety to the C6 of the Glc residue of Suc through a β(2-6) bond. This results in the synthesis of neokestose, the starting brick for the build-up of neofructan series. Neofructans can be further divided into neoinulins, neolevans, and agavins [46] (Figure 1). Besides their role as reserve compounds, fructans are recognized players in plant adaptation responses to abiotic stress [47,48,49]. Their contribution to physiological adjustments during drought [50,51] and cold stress [52,53] is well described, involving mechanisms such as osmotic protection of cellular components. Besides their relevance in abiotic stress adaptation, an emerging role for fructans under biotic challenges has been proposed [54,55]. Fructan accumulation *in vivo* was linked to pathogen resistance in plant species such as agave and wheat [54,56] and exogenous application of inulins extracted from burdock (*Arctium lappa*) roots (burdock fructooligosaccharides (BFOs)) proved to be effective in boosting plant resistance towards biotic attacks in pathosystems such as *Nicotiana tabacum*-tobacco mosaic virus and *Cucumis sativus*-*Colletotrichium orbiculare* [57,58]. Thus, the use of fructans as priming agents in the biocontrol context represents an attractive possibility. The goal of this study was to test the potential of inulin fructans to induce resistance in the leafy vegetable lettuce (*Lactuca sativa*) against the model necrotrophic fungus *Botrytis cinerea*, able to infect more than 200 plant species [59], and to study the underlying mechanisms. 

## 2. Results

### 2.1. Burdock Inulin Is Effective in Protecting Lettuce Leaves against Botrytis

First, the immunostimulatory activity of BFOs was tested in our pathosystem, composed of lettuce (*Lactuca sativa* var. Gisela) and *Botrytis cinerea* strain B05.10 [60]. The same concentration reported from the previous papers (5 g·L^−1^) was used to spray 45 day-old lettuce plants and inoculate them 3 days after priming with *Botrytis*. To confirm that the observed effect is not associated with impurities (small soluble sugars Glc, Fru, and Suc and the smallest DP (degree of polymerization) 3–6 inulin type fructans), present in our fructan formulation, BFOs priming was compared with the subsequent controls: fully hydrolyzed BFOs (H BFOs), BFOs dialyzed with a 1 kDa-cutoff membrane (which allows exchange of fructan oligosaccharides up to DP = 6) (D BFOs), hydrolyzed and dialyzed BFOs (HD BFOs). As a positive control, oligogalacturonides (OGs) was selected, which is a well-characterized DAMPs (Damage-Associated Molecular Patterns) [36] at a concentration of 0.5 g·L^−1^. Conductivity tests estimated the molarity of a 5 g·L^−1^ BFOs solution between 50 and 55 mM. Thus, to take possible osmotic effects into account, sorbitol 50 mM was used as osmotic control. After priming, fully-grown sprayed leaves were detached and inoculated. Figure 2 shows a significant decrease in disease symptoms in leaves treated with BFOs, D BFOs, and OGs (Figure 2A,B). These data confirm the priming efficacy of BFOs in our pathosystem, excluding any artifacts due to osmotic or Fru-dependent effects. Furthermore, application of pure DP > 6 BFOs resulted in an efficient priming, demonstrating that the protective effect could not be attributed to small soluble sugar impurities. In addition, we proved for the first time that OGs are effective defense elicitors on lettuce plants. It was also tested whether exogenous application of the common soluble sugars Suc, Glc, and Fru at 50 mM had an effect on the defense response against *Botrytis*, but no positive results were found (Appendix A). 

### 2.2. Chicory-Derived Inulin Is Effective in Inducing Protection in the Lettuce-Botrytis cinerea Pathosystem

To understand whether the immunostimulatory effect of BFOs is source-specific or shared by inulin-type fructans extracted from other sources, BFOs priming efficiency of commercially available inulin (Sigma) derived from chicory (*Cichorium intybus*) at similar concentrations. Figure 3A compares the sugar profiles of BFOs extracted from burdock and commercial inulin derived from chicory. Disease scoring results revealed that commercial chicory-derived inulin priming was still efficient in significantly decreasing the disease symptoms in our pathosystem (Figure 3B). These data confirm that inulin-type fructans are effective priming agents in this pathosystem regardless of their origin. Decreasing the inulin concentration from 5 to 1 g·L^−1^ still led to significantly decreased disease symptoms (Figure 3C). Comparing the 1 and 5 g·L^−1^ treatments, no obvious differences were observed in disease symptoms. *Botrytis* growth on PDA plates supplemented with different inulin concentrations was tested as well, without finding differences between the treatments (Appendix A). This result suggests that the inulin effect is plant-mediated. For subsequent experiments, chicory inulin at 1 g·L^−1^ was used. 

### 2.3. Inulin Treatment Induces H_2_O_2_ Accumulation

Histochemical measurements of H_2_O_2_ accumulation were performed in primed and infected leaves using the DAB (3,3′-diaminobenzidine) assay [61]. Leaves were sampled 3 h post priming (3 hPP), 1 day post priming (1 DPP), 0 days post inoculation (or 3 days post priming) (0 DPI), and 1 day post inoculation (1 DPI). To discriminate between leaf detachment and inoculation effects, an additional control without *Botrytis* inoculation (uninfected) was sampled 1 DPI. Results are shown in Figure 4 and more representative pictures are shown in Appendix A. Already 3 h after spraying OGs and inulin, a significant H_2_O_2_ accumulation was detected and this trend was maintained at 1 DPP, indicating a steady production of H_2_O_2_ over the first 24 h. At 3 DPP, H_2_O_2_ levels in inulin samples decreased compared to the mock treatment, whereas H_2_O_2_ levels of OGs samples remained steady (Figure 4A). After infection, a drastic increase in H_2_O_2_ levels was visible in both OGs and inulin-treated leaves. An increase in H_2_O_2_ content in the mock sample was also observed compared to the time-point before inoculation, however this effect was likely due to leaf detachment, because a similar H_2_O_2_ level was detected in the uninfected control (Figure 4A). 

### 2.4. Inulin-Enhanced Plant Defense Requires Ethylene Signaling

Since in lettuce, ethylene signaling was shown to be activated in response to *Botrytis* infection [60], we hypothesized that ethylene signaling could be involved in basal and elicitor-induced protection to *Botrytis* in lettuce. To test this, we treated mock, OGs, and inulin-primed plants with 10 ppm 1-MCP (1-methylcyclopropene), a competitive inhibitor of ethylene signaling [62], for 24 h before inoculation and compared disease scoring results with the ones from untreated plants. Results revealed that both inulin and OGs were no longer able to decrease disease symptoms after a 1-MCP treatment (Figure 5B), indicating that a functional ethylene signaling pathway is required to enhance the defense response after priming with these compounds. Interestingly, no statistical difference was detected by comparing mock treatments with and without 1-MCP application (*p* = 0.904, Mann–Whitney *u*-test), suggesting that ethylene signaling is not required for basal resistance in mature lettuce leaves against *Botrytis*. To test whether 1-MCP application may have a direct effect on *Botrytis* growth and sporulation, we fumigated PDA plates inoculated with *Botrytis* with 10 ppm 1-MCP and looked for differences in mycelial growth compared to untreated plates after 7 and 12 days. No differences were detected at either time-point tested (Figure 5C), indicating that 1-MCP does not affect *Botrytis* growth on the plate. We tried the same experiment using 10 µM AVG (2-aminoethoxyvinyl glycine), an ethylene biosynthesis inhibitor which blocks synthesis of 1-aminocyclopropane-1-carboxylic acid (ACC) [63]. We obtained a similar reduction of OGs and inulin-triggered protection, although results were less pronounced when compared to 1-MCP data (Appendix A). This suggests that ethylene synthesis plays a role in inulin and OGs-induced protection but is not required for basal resistance in mature lettuce leaves against *Botrytis*. 

### 2.5. Oligogalacturonides (OGs) Treatment and 1-Methylcyclopropene (1-MCP) Application Induce Changes in Soluble Sugar Content, and Have a Major Impact on Suc Accumulation

Fru, Glc, and Suc content of leaf samples collected immediately before inoculation (0 DPI or 3 DPP) and 1 day post inoculation (1 DPI) were analyzed. To properly take into account the effect of priming, samples taken immediately before the priming event (0 days post priming, 0 DPP) were also collected. To distinguish between detachment and pathogen effects, uninfected control leaves were sampled at 1 DPI. Results are summarized in Figure 6, showing that Suc levels increased after OGs priming (Figure 6A). Interestingly, during infection a significant decrease in Suc content was registered for the same treatment (Figure 6B). Regarding the 1-MCP effect on Suc levels, a significant drop in all the treatments was detected before and after inoculation, with the only exception of OGs samples at 1 DPI (Figure 6B). These data suggest a strong negative effect of 1-MCP on Suc accumulation. Similar to Suc, Glc levels were positively affected by OGs priming, and the 1-MCP application lowered Glc levels in all treatments, although not significantly in inulin samples (Figure 6C). Higher Glc levels were registered in uninfected samples as compared to infected samples, suggesting an increase in Glc use in the presence of the pathogen. However, we do not know whether this observation can be ascribed to direct nutrient utilization from the host or is due to the pathogen. 1-MCP negatively affected only inulin samples at this time-point (Figure 6D). Regarding Fru levels, again a positive effect of OGs priming was observed, while the 1-MCP treatment had no significant effect at this stage (Figure 6E). A strong increase in Fru levels was detected in all infected samples, including the uninfected control, which might suggest that detachment induces Fru accumulation in lettuce leaves (Figure 6F). As for Glc, 1-MCP treatments significantly decreased Fru levels only for inulin samples (Figure 6F). Our data indicate an overall negative effect of 1-MCP on the Suc pool, a minor effect on the hexose pool, and a particularly drastic effect on inulin-primed samples. Regarding priming treatments, OGs stimulate an accumulation of soluble sugars after priming.

### 2.6. Effect of Priming and 1-MCP on Acidic Invertase Activities 

Total activity of soluble (VI, Vacuolar Invertase) and insoluble (CWI, Cell Wall Invertase) acidic invertases was measured in our samples in order to assess their involvement in priming and response to *Botrytis*. Results collected for CWI activity show that priming treatments had no effect on activity levels of invertases, but their activity was enhanced by the 1-MCP treatment in all samples, although with a smaller magnitude in the case of CWI after inulin priming (Figure 7A). At 1 DPI, the 1-MCP effect appears to become even stronger, independently from the priming treatment considered. No differences between control priming treatments (mock, OGs, inulin) were observed (Figure 7B). Importantly, activity levels of CWI in uninfected samples raised when compared to all the other infected samples (Figure 7B). This suggests that pathogen perception at this early stage triggers a decrease in CWI activity, contrary to what was expected. VI activities at 0 DPI followed substantially the same trend of the corresponding CWI data, with a strong increase in activity after the 1-MCP treatment for all priming agents compared to the control treatment. In contradiction to CWI activity, this increase in VI activity was more prominent in the inulin-treated samples (Figure 7C). At 1 DPI, VI activity increased in mock samples to a similar extent to what was observed for CWI (Figure 7D). A significant decrease in VI activity was noticed in inulin and uninfected samples when compared to the mock. Interestingly, the 1-MCP application resulted in a completely opposite outcome of VI activity with respect to the CWI 1 DPI data: where 1-MCP stimulates CWI activity 1 DPI, it does not do so for VI for the mock, OGs and inulin primed samples. However, a slight increase of VI activity of the uninfected samples treated with 1-MCP was recorded (Figure 7D). Our data suggests that ethylene signaling has a negative effect on both CWI and VI post priming, but during infection this repression is lost for VI but not for CWI.

### 2.7. γ-Aminobutyric Acid (GABA) Accumulates in Inulin-Primed Leaves after Botrytis Infection

γ-Aminobutyric acid (GABA) levels in leaf samples from the previous 1-MCP experiment were quantified. Results obtained at 0 DPI did not show differences between priming treatments for both the 1-MCP treated and untreated leaves, although a decreasing trend following 1-MCP treatment can be recognized in this respect (Figure 8A). Data collected at 1 DPI showed an increase in GABA levels for inulin-treated samples (Figure 8B). Interestingly, 1-MCP only had a negative effect on GABA content in the OG and inulin primed leaves (Figure 8B). These results suggest that ethylene signaling is required to induce GABA accumulation after priming and during subsequent infection. Proline (Pro) levels in our samples were assessed as well, and results at 0 DPI indicated no significant differences between priming treatments or the 1-MCP application (Figure 8C). At 1 DPI, a significant increase in mock-primed samples was observed when compared to uninfected control, indicating that *Botrytis* infection triggered Pro accumulation (Figure 8D). 1-MCP treatment did not result in significant differences in Pro accumulation compared to control treatments, although lowered Pro values in OGs and inulin primed samples were detected. Furthermore, 1-MCP treated mock samples had significantly higher Pro levels compared to OGs, inulin and uninfected samples treated with 1-MCP (Figure 8D). These results suggest a stimulation of Pro production during infection. There does not seem to be a clear effect of priming on Pro levels before and after infection. The role of 1-MCP is not clear, as it only lowered Pro levels at 1 DPI. 

## 3. Discussion

### 3.1. Fructan and OGs Elicit Defense Responses against Botrytis in Lettuce Leaves

Many efforts are being made to develop alternatives to the use of toxic pesticides. Priming with natural products is a valuable option in this respect. The possibility of using sustainable, cheap, and easily available compounds like carbohydrates is receiving increasing attention [25], and illustrates the applicability of the Sweet Immunity concept in horticultural and postharvest practices [64,65]. We demonstrate that inulin-type fructans can successfully induce defense responses against *Botrytis* in lettuce (Figure 2). Our data show that the immunostimulatory effect can be associated with the specific saccharidic nature of the inulins (DP ≥ 6) (Figure 2). Both BFOs and chicory-derived inulins can enhance defense responses against *Botrytis* in lettuce, suggesting that inulin-type fructans have immunostimulating properties regardless of their origin. This further reinforces the conclusion that the ability to limit disease progression can be ascribed to the structural features of inulin (Figure 3). Lettuce is able to synthetize inulin-type fructans in stems and roots in modest amounts, but not in leaves [66]. Although there is no known fructan receptor and/or signaling pathway identified in plants, work in T84 human intestinal epithelial cell monolayers showed that inulin is able to bind and activate Toll-Like Receptor 2 (TLR2) and to a lesser extent TLR4, TLR5, TLR7, and TLR8 [67,68]. Such studies lead us to formulate the hypothesis that fructans may be perceived in plants as well, in the role of MAMPs or DAMPs [69] by a so far unknown sensor (TLR homologues are not present in plant genomes). Another option could be that the biological effects of fructans would involve their well-described interaction with membranes [70]. OGs are prototypical DAMPs able to elicit defenses against *Botrytis cinerea* in *Arabidopsis* and grape [71,72]. They may prime defenses upon cell wall damage [36]. We proved for the first time the effectiveness of OGs in inducing disease protection in lettuce (Figure 4) and optimized the best dosage (Appendix A) to induce protection, and used them as positive control.

### 3.2. H_2_O_2_ Accumulates Following Fructans and OGs Treatments

A common feature of plant responses to pathogens is the production and accumulation of reactive oxygen species (ROS) [73,74], and induction of ROS accumulation is widely documented in leaf tissues subjected to abiotic stress and exposed to DAMPs and MAMPs [71,75]. BFOs are able to induce H_2_O_2_ accumulation in tobacco leaves, with a peak at 6 h after treatment [57]. Our data showed a modest, but significant oxidative burst for inulin and OGs at 3 h after spraying and this level remained high for the subsequent 72 h, although for inulin we could not detect significant differences anymore at 72 h (Figure 4). Overall, time-dependent ROS dynamics may greatly depend on the dosage and species considered [71,76]. Altogether, the increase in H_2_O_2_ levels after inulin priming might suggest it triggers a ROS-mediated plant defense response. As expected, ROS levels increase much more when the pathogen comes into play, as compared to pre-inoculation time-points (Figure 4). In general, ROS generation is associated with SA signaling, which is effective against biotrophs, whereas necrotrophs usually take advantage of it [77]. In *Arabidopsis*, OGs elicit an oxidative burst, but this burst is not required for the OG-induced resistance to *B. cinerea* [78]. Thus, it remains possible that the H_2_O_2_ induction observed in lettuce treated with OGs (Figure 4) is not a major factor against *Botrytis*. Therefore, it is difficult to say if this is also the case for the increase in H_2_O_2_ levels observed in inulin-treated leaves, and whether it is related to disease limitation or not. However, it was shown that timely and localized ROS induction at the early stages of the plant-pathogen interaction can be a resistance factor against a necrotroph such as *Botrytis* in tomato and bean [79,80]. This may be the case for our pathosystem as well, but further research is required to clarify this. 

### 3.3. Ethylene Is Involved in OGs and Fructan Priming in Lettuce

Ethylene and JA are the most important phytohormones in modulating resistance against necrotrophs [14]. Ethylene was shown to be required for OGs-mediated *Botrytis* resistance in *Arabidopsis* [11], and our ethylene synthesis and signaling inhibitor (1-MCP and AVG) treatments suggest a similar case for lettuce (Figure 5, and Appendix A). It is also the first time that ethylene synthesis and signaling were studied in combination with fructan treatment. The fact that inulin-triggered disease limitation also seems to rely on ethylene signaling and synthesis is a novel finding. Perhaps the most interesting observation is that ethylene signaling and synthesis are predominantly playing an important role in primed leaves, because a 1-MCP and AVG treatment does not increase susceptibility in mock-treated leaves (Figure 5, and Appendix A). These data suggest that ethylene is not involved in basal immunity in water-treated lettuce leaves. Ethylene is known to be a resistance factor in leaves of several plant species against *Botrytis* (*Arabidopsis*, artemisia, tomato) [11,81,82]. However, it was also demonstrated that a 1-MCP application did not alter *Botrytis* resistance in pepper leaves in the absence of priming [83]. Taken together, our results add another layer of complexity regarding the role of ethylene during plant-necrotroph interactions, with many possible variations depending on the pathosystem as well as the stage of infection [20]. JA involvement in *Botrytis* resistance was demonstrated in species such as *Arabidopsis* and tomato [84,85]. Although we have not studied JA in our pathosystem, further research to clarify its involvement and possible interaction with the ethylene pathway in lettuce is warranted.

### 3.4. OGs Induce Accumulation of Soluble Sugars before Botrytis Inoculation

The changes in soluble sugars content after priming suggest that the OGs treatment triggers accumulation of Suc, Glc, and Fru in lettuce leaves (Figure 6A,C,E). To the best of our knowledge, it is the first time that soluble sugar dynamics are recorded after application of OGs and subsequent infection. On the other hand, inulin priming does not result in significant changes in the sugar pools before inoculation (Figure 6A,C,E), while the level of hexoses increased after infection (Figure 6D,F). These data reveal clear differences in how exogenous applications of OGs and inulins influence sugar metabolism. Glc and Suc, but not Fru, levels dropped drastically in OGs samples at 1 DPI (Figure 6B,D,F). This may indicate an increase in the use of Glc moieties after pathogen perception in leaves pretreated with OGs. It will be interesting to address future studies on how OGs application impacts apoplastic sugar concentrations.

### 3.5. Suc Dynamics Are Modulated by Ethylene Signaling

The interaction between ethylene and sugar metabolism represents an interesting topic in present day plant physiology, although the regulatory mechanisms may greatly depend on the source and sink status of the organs, the developmental stage, and on the species studied [86,87]. On the one hand, ethylene stimulates senescence in aging leaves and ripening in fruits, while on the other hand, it is essential for normal plant development, and for many physiological processes, including photosynthesis [86,87]. We observed a strong negative effect of 1-MCP on the Suc pool both before and after inoculation, independently of priming (Figure 6A,B). This result indicates that an intact ethylene signaling pathway is required to maintain physiological Suc levels in mature lettuce leaves. Given the importance of Suc as a building block for antimicrobial compounds and as a signaling molecule in plant defense [38,88,89], a decrease in Suc levels may be one of the explanations for the increased susceptibility observed in OGs and inulin treated leaves after the 1-MCP application. This leaves us with a question: why is such susceptibility increase not detected in mock-treated plants? It is also worthwhile to note that 1-MCP strongly decreases all soluble sugars in inulin primed samples after infection, suggesting that inulin priming relies on ethylene-dependent sugar homeostasis (Figure 7B,D,F). Notably, 1-MCP does not have a major influence on the Fru pool (Figure 6E,F). In contrast to Fru, Glc levels can drop fast, because it is rapidly channeled into the OPPP pathway by glucose 6-phosphate dehydrogenase (G6PD) activity [90]. This pathway provides NADPH (Nicotinamide Adenine Dinucleotide Phosphate) for reductive biosynthesis and maintenance of the cellular redox state, crucial during defense responses [91]. 

### 3.6. Ethylene Differentially Controls Cell Wall and Vacuolar Invertase Activities

Invertases can modulate Suc/hexose balances. As such, they can greatly influence sugar metabolism and sugar signaling events. Our data on acidic invertase activities revealed no significant changes following priming treatments (Figure 7A,C), although we cannot rule out the possibility that we missed an earlier time window of invertase (in)activation. There is an extensive body of evidence that CWI activity increases following pathogen perception in different pathosystems [92,93,94], suggesting that CWI contributes to pathogen resistance [95]. Unexpectedly, we found an increase in CWI activity in uninfected samples, as compared to the other treatments at 1 DPI (Figure 7B). A first interpretation is that the situation in lettuce may be different as compared to the above-mentioned species. Another possible explanation could lie in the sampling. Whereas most studies used a full-leaf sampling, taking into account the main vein, our sampling was much narrower around the lesion and possibly not representative for a full-leaf context. 

Ethylene negatively affects CWI activity in cell suspensions of *Chenopodium rubrum* [96] and increased CWI activity is observed in *Arabidopsis ein4* mutants [97], suggesting that ethylene suppresses CWI activity. Accordingly, the 1-MCP application in our work led to a drastic increase in CWI activity (Figure 8B). However, 1-MCP seems unable to release the ethylene suppression of CWI activity in uninfected controls. Lowered CWI activities at the very early stages of infection may be advantageous to limit hexoses in the apoplastic environment, counteracting the spreading of *Botrytis* [98]. Accordingly, cases of decreased infection symptoms development following CWI repression are documented [99]. Such a mechanism may be advantageous for the plant, especially if combined with accelerated Suc uptake and intracellular processing by VI. Accordingly, VI activities are significantly higher in water-primed and infected leaves as compared to uninfected leaves (Figure 7D), while the opposite reaction can be observed for CWI. Interestingly, inulin seems to counteract this effect on VI (Figure 7D). It is not clear whether VI has a positive or negative role for the plant in our pathosystem, given that it is repressed by inulin but not affected by OGs at 1 DPI (Figure 7D). Moreover, the 1-MCP treatment enhanced VI activity at 0 DPI in all treatments, but had no effect on infected samples at 1 DPI (Figure 7C,D). These data suggest that ethylene plays a negative role in regulating both CWI and VI [97], but after pathogen perception this control mechanism seems to be uncoupled. It remains to be tested whether such uncoupling is due to the pathogens’ capacity to change soluble sugars availability, by direct uptake or by using dedicated effectors.

### 3.7. Inulin Treatments Induce GABA Accumulation During Botrytis Infection

GABA plays a central role at the interface of C and N metabolism [100]. In the tomato-*Botrytis* pathosystem, the GABA shunt was shown to be critical in replenishing the TCA (Tricarboxylic Acid) cycle and eventually maintaining cell viability during *Botrytis* infection in ABA-deficient *sitiens* mutants of tomato [101]. The increase in GABA levels in inulin-treated leaves after infection may help to explain the observed enhanced defense responses (Figure 8B) in view of the model proposed by Seifi and colleagues [101]. In line with their work, GABA and H_2_O_2_ may fulfil complementary roles, maintaining cell viability on the one hand and preventing rapid spreading of the pathogen on the other hand [101]. Given that inulin priming induces both GABA and H_2_O_2_ accumulation, it will be interesting to dedicate future research to the link between ABA and fructan priming/perception. Interestingly, 1-MCP impairs GABA accumulation in OGs and inulin treated leaves after infection (Figure 8B), indicating that ethylene signaling mediates GABA accumulation after a *Botrytis* infection. To the best of our knowledge, no reports are available on GABA induction by ethylene in the context of biotic stress. However, it was shown that GABA can induce ethylene synthesis in sunflower [102], raising the hypothesis of a positive feedback mechanism between these two signaling compounds following priming events. 

Pro accumulation and metabolism is known to contribute to plant resistance against biotrophs [103], and Pro dehydrogenase mutants were shown to have impaired resistance against both *Pseudomonas syringae* and *Botrytis cinerea* [101,104]. *Botrytis* infection stimulates Pro accumulation in lettuce (Figure 8D), but the lack of significant differences between mock and primed samples suggest that Pro concentration does not play a significant role in our pathosystem, at least at the early stages of infection. 

## 4. Materials and Methods

### 4.1. Biological Material

Lettuce plants (*Lactuca sativa* L. cv. Gisela) were grown in a Conviron^®^ chamber with a temperature of 21/16 °C (day/night), 12 h/12 h day/night light regime, light intensity of 170 µmol·m^−2^·s^−1^, and 80% RH (Relative Humidity). For experiments, fully mature leaves from 45 days old plants were used. *Botrytis cinerea* B05.10 strain was used in our trials [60], provided by Dr. Barbara De Coninck (KU Leuven, Division of Crop Biotechnics, Leuven, Belgium), and cultivated on potato dextrose agar plates (PDA) at 18 °C. Before inoculations, spores were harvested from plates and diluted in half-strength potato dextrose broth (PDB) to a final concentration of 5 × 10^5^ spores mL^−1^ with the help of a stereomicroscope Olympus 3000 (Olympus, Tokyo, Japan).

### 4.2. Priming and Infection Assays

For priming treatments, four mature leaves from each plant were selected, and sprayed with different solutions. From 4 to 6 plants per treatment were used. All the solutions were prepared in Milli Q water supplemented with 0.0001% Tween 20 as surfactant. Three days after priming, two 5 µL drops of the above mentioned *Botrytis* solution were placed on detached leaves (Figure 9). Inoculated leaves were placed in Petri plates of 15 cm diameter, lined with wet tissue paper and sealed with parafilm. Plates were placed in an incubator at 18 °C for 4 days before disease scoring. Necrotic lesion area was calculated by using ImageJ program (https://imagej.nih.gov/ij/) as described [72]. Lesions were then assigned to one of the 6 severity categories as defined in Table 1. 

### 4.3. H_2_O_2_ Assay

Detection of H_2_O_2_ was performed according to Daudi et al. [61], with minor modifications. Leaves were divided into 4 segments in order to facilitate infiltration. Briefly, leaf segments were infiltrated with DAB solution at 200 mBar until tissues were completely infiltrated, and then incubated in dark for 4 h. Subsequently, leaves were de-stained in an ethanol:acetic acid:glycerol 3:1:1 solution and photographed on a white background. ROS quantification was performed with a pixel analysis method by converting images to grey scale, as described [105]. 

### 4.4. 1-MCP and AVG Treatments

Anti-ethylene treatments were done with an ethylene perception inhibitor (1-MCP; binds to the ethylene receptors) and an ethylene biosynthesis inhibitor (AVG; blocks ACC-synthase activity) 24 h before inoculation. Lettuce plants were placed in airtight sealed Plexiglas boxes and administered with 10 ppm 1-MCP (1-methylcyclopropene; a kind gift from AgroFresh, Philadelphia, PA, USA) for 24 h. Dry 1-MCP powder was wetted with water after sealing the boxes by gently shaking. For the AVG treatments, plants were sprayed 4 h before inoculation with 10 µM AVG (aminoethoxyvinylglycine; Sigma, St. Louis, MO, USA) supplemented with 0.0001% Tween 20 as surfactant. Leaves were detached 24 h after the 1-MCP or AVG treatment and prior to the inoculation treatment.

### 4.5. Carbohydrate Extraction, Processing, and Analysis

To extract and quantify Glc, Fru and Suc, a circular shape (3.5 cm diameter) of leaf material with an approximate weight of 200 mg was taken from primed and infected leaves. In the case of infected leaves, a circular leaf disc (3.5 cm diameter) surrounding the lesion was sampled. This material was ground in liquid nitrogen and then heated at 90 °C in 0.5 mL water, and subsequently passed through Dowex^®^ anion and cation exchange resins. Rhamnose 20 µM was added as internal standard. The sample was then analyzed by HPAEC-IPAD Dionex 3000 (Thermofisher, Whaltam, MA, USA) according to Shiomi et al., [106]. 

BFOs were extracted from freeze-dried burdock (*Arctium lappa*) roots according to the protocol described by Hao et al. [107] with minor modifications. For enzymatic hydrolysis treatments, we used a -fructan 1-exohydrolase from chicory [108] immobilized on a Concanavalin-A Sephadex resin (GE Healthcare, Chicago, IL, USA). BFOs were dialyzed with 1 kDa cutoff Spectra/Por membrane (Repligen, Waltham, MA, USA).

OGs were extracted from polygalacturonic acid sodium salt (Sigma). 800 mg of powder were incubated with 10 mg of pectinase extracted from *Aspergillus niger* (0.71 units/mg, Polylab, Kundli, India) in 60 mL NaAc buffer pH 5 for 60 min at 40 °C. The solution was then boiled for 8 min at 90 °C and centrifuged (20 min, 40,000× *g*). The resulting supernatant was diluted in 40 mL acetone and centrifuged (15 min, 14,000× *g*). This step was repeated twice. Finally, the obtained pellet was washed two times in 80% acetone and dried with the use of a liofilizator (LSL Secfroid, Aclens, Switzerland).

### 4.6. Acidic Invertase Activity Assays

For VI and CWI activity assays, 50 mg of grounded lettuce leaf material was dissolved in 150 µL of extraction buffer 50 mM NaAc buffer at pH 5, with 10 mM NaHSO_3_, 0.01% Polyclar and 2 mM β-mercaptoethanol (β-ME). 1 µL of 200 mM phenylmethylsulfonyl fluoride (PMSF) was added to each sample. The homogenized solution was centrifuged at 13,000× *g* for 15 min, and pellet (CWI-enriched) and supernatant (VI-enriched) were separated. De-salting and invertase activity measurements were carried out as described [109].

### 4.7. GABA Extraction and Analysis

GABA was extracted from 50 mg of ground lettuce leaf in 100 µL water and subsequently sonicated for 10 min. Samples were then centrifuged at 13,000× *g* for 5 min. 45 µL of supernatant was diluted with 30 µL of a 25 mM NorValine solution (used as internal standard). GABA was then injected on a reverse HPLC system (Shimadzu, Kyoto, Japan) and derivatized before column injection as *O*-Phtalaldehyde and separated on a YMC Triart C18 column with the use of buffer “A” (50 mM KH_2_PO_4_ pH 6.5, 0.7% *v*/*v* Tetrahydrofurane) and buffer “B” (Acetonitrile:MeOH:water 45:40:15) with the following gradient: from 0 to 6 min = 96% A 4% B; from 6 to 18 min = 92% A 8% B; from 18 to 32 min = 85% A 15% B; from 32 to 50 min = 67% A 33% B; from 50 to 53 min = 100% B. GABA was detected with a fluorescence detector at λ_ex_ = 230 nm and λ_em_ = 450 nm.

### 4.8. Proline Extraction and Analysis

Pro was extracted from 50 mg of ground lettuce leaf with 100 µL water and subsequently sonicated for 10 min. Samples were then centrifuged at 13,000× *g* for 5 min. The supernatant was diluted 10 times in water containing 20 µM serine (used as internal standard). The sample was then vortexed and centrifuged at 13,000× *g* for 5 min, transferred to a vial with a conical glass insert and analyzed by HPAEC-IPAD (Thermofisher, USA) in order to determine the Pro content after separation on a CarboPac^®^ PA100 anion exchange column. The flow rate was 0.25 mL·min^−1^. Pro was eluted with the following gradient: 9 mM NaOH from 0 min to 15 min; 0.5 M NaAc from 15 to 17 min; 90 mM NaOH from 17 to 23 min. Quantification was performed on the peak areas with the external standard, consisting of 20 µM Pro. 

### 4.9. Statistical Analysis

Disease scoring data were analyzed by using two-tailed Mann–Whitney *u*-test for categorized data. Other data were analyzed by using one-way ANOVA followed by Tukey’s post-hoc test, or two-tailed Student’s *t*-test when appropriate. 

## 5. Conclusions

In this study, we show that inulin-type fructans are promising priming agents in the context of disease control, an observation that should stimulate research on fructans and on other carbohydrates as potential substitutes for pesticides use in field and postharvest applications [27,28]. In this sense, an important future task will be to understand which natural priming agents are the most efficient disease suppressors in specific crops, as well as exploring possibilities of inducing cross-tolerance towards abiotic and biotic stresses. The observed changes in H_2_O_2_ accumulation, sugar metabolism, and GABA dynamics, as well as the clear connections with ethylene signaling, opens new research avenues to address the role of fructans in the sweet immunity context.

## Figures and Tables

**Figure 1 ijms-20-01052-f001:**
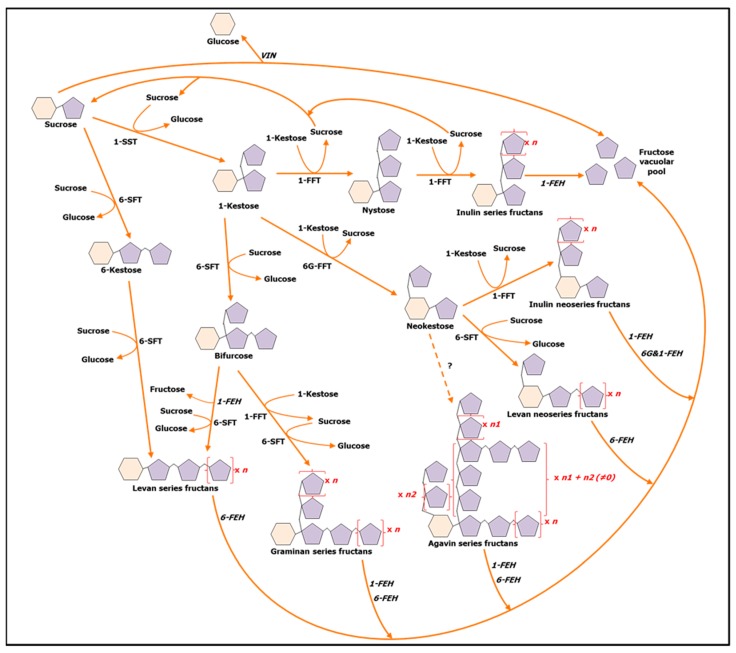
Schematic representation of fructan metabolism in plants. FEH = Fructan Exohydrolase; FFT = Fructan:Fructan Fructosyltransferase; SFT = Sucrose:Fructan Fructosyltransferase; SST = Sucrose:Sucrose Fructosyltransferase; VIN = Vacuolar Invertase. 1- and 6- prefixes of enzymes refer to the position of the C of the Fru moiety that is attacked by the enzyme. The 6G- prefix indicates that the enzyme acts on the C6 of the Glc moiety. All the described reactions take place in the vacuole [45,46]. The biosynthetic pathway of agavins are still under investigation. Although fructans are believed to be stored in the vacuole, their presence in the apoplast suggested a possible vesicular-mediated transport between the vacuole and the apoplast [47].

**Figure 2 ijms-20-01052-f002:**
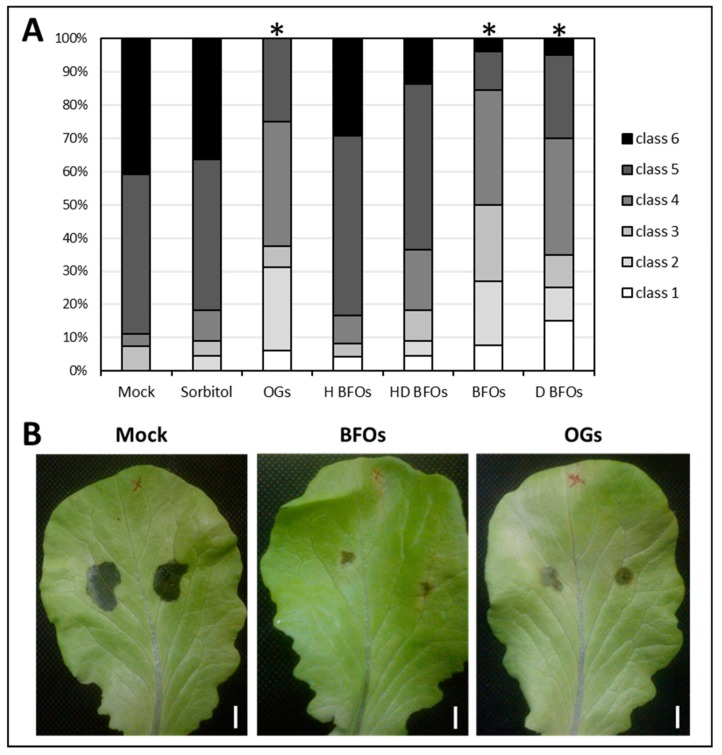
(**A**) Stacked-column chart illustrating severity class distribution resulting from the burdock fructooligosaccharides (BFOs) specificity test. Asterisks indicate significance against mock (water) at *p* < 0.01 according to non-parametrical, two-tailed, Mann–Whitney *u*-test, *n* ≥ 16. This experiment was repeated three times with consistent results; (**B**) Representative pictures of *Botrytis* lesions on lettuce leaves treated with water (mock), BFOs 5 g·L^−1^ and oligogalacturonides (OGs) 0.5 g·L^−1^ at 4 DPI (days post inoculation). Bars = 1 cm.

**Figure 3 ijms-20-01052-f003:**
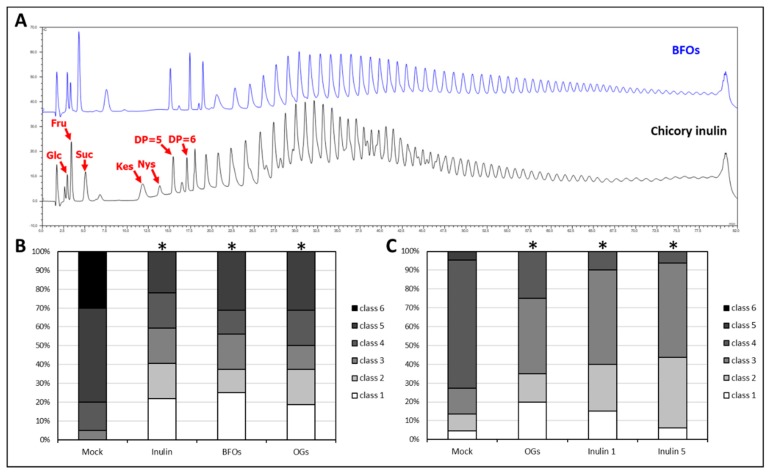
(**A**) Comparison between chromatograms resulting from injection of BFOs and chicory inulin (HPAEC-IPAD). *Y* axis = detector response in nanoCoulomb; *X* axis = elution time in min; Glc = glucose; Fru = fructose; Suc = sucrose; Kes = 1-kestose; Nys = nystose; (**B**) Results of the disease scoring experiment comparing BFOs and chicory inulin priming efficiency. Asterisks indicate significance against mock (water) at *p* < 0.01 according to non-parametrical, two-tailed, Mann–Whitney *u*-test. This experiment was repeated three times with consistent results; (**C**) Results of the disease scoring experiment comparing chicory inulin at 1 g·L^−1^ (inulin 1) and the same solution at 5 g·L^−1^ (inulin 5) priming efficiency. Asterisks indicate significance against mock at *p* < 0.01 according to Mann–Whitney *u*-test. This experiment was repeated three times with consistent results.

**Figure 4 ijms-20-01052-f004:**
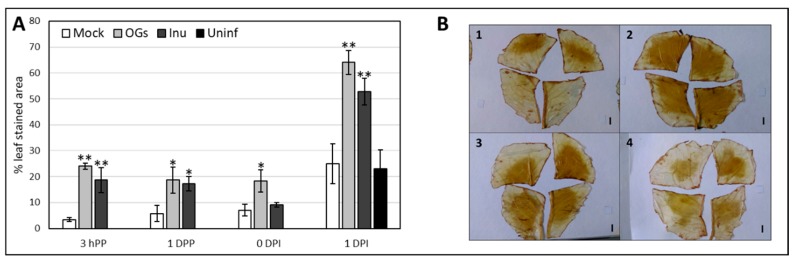
Induction of H_2_O_2_ accumulation by OGs and inulin priming. Mock = water-primed leaves; OGs = OGs-primed leaves; Inu = chicory inulin-primed leaves; Uninf = water-primed leaves inoculated with a mock solution (1/2 strength Potato dextrose broth). (**A**) The bar charts illustrate the percentage of DAB (3,3′-diaminobenzidine) stained leaf area in mock-treated and primed leaves at different time-points. 3 hPP= 3 h post priming; 1 DPP = 1 day post priming; 0 DPI = 0 days post inoculation; 1 DPI = 1 day post inoculation. Asterisks indicate statistically significant differences between treatments and mock for each time-point (* *p* < 0.05, ** *p* < 0.01) according to Students *t*-test. The experiment was repeated twice with consistent results; (**B**) Representative pictures of DAB-stained leaves collected at 1 DPI. (1) Mock (2) OGs (3) Inulin (4) Uninfected. Scale bars = 1 cm.

**Figure 5 ijms-20-01052-f005:**
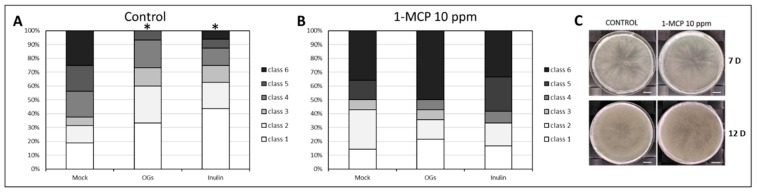
Effect of 1-methylcyclopropene (1-MCP) treatment on basal and OGs- and inulin-induced protection against *Botrytis cinerea*. (**A**) Disease scoring results of primed leaves not treated with 1-MCP; (**B**) Disease scoring results of primed leaves treated for 24 h with 10 ppm 1-MCP before pathogen inoculation. Asterisks indicate significance against mock (*p* < 0.05, Mann–Whitney *u*-test). This experiment was repeated three times with consistent results; (**C**) Representative pictures of *Botrytis cinerea* growth assay on PDA plates after a treatment with 10 ppm 1-MCP, with pictures taken 7 days (7 D) and 12 days (12 D) after plate inoculation. Bars = 1 cm.

**Figure 6 ijms-20-01052-f006:**
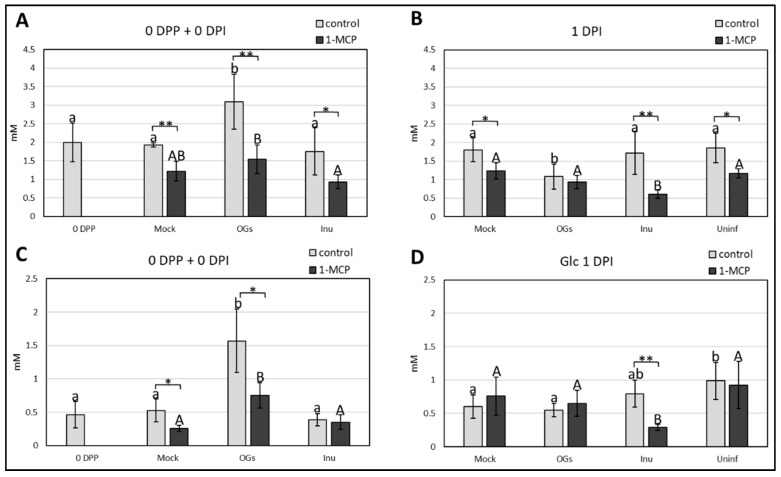
Soluble sugar analysis from leaf samples collected from the 1-MCP experiment. Mock = water-primed leaves; OGs = OGs-primed leaves; Inu = chicory inulin-primed leaves; Uninf = Water-primed leaves inoculated with a mock solution (1/2 strength Potato dextrose broth). (**A**) Suc quantification at 0 DPP (before priming = 0 DPP) and 0 DPI (3 days after priming and before inoculation); (**B**) Suc quantification at 1 DPI; (**C**) Glc quantification at 0 DPP and 0 DPI; (**D**) Glc quantification at 1 DPI; (**E**) Fru quantification at 0 DPP and 0 DPI; (**F**) Fru quantification at 1 DPI. Bars marked with same letter(s) are not significant between each other, bars that have no letters in common indicate significance between treatments (*p* < 0.05, one-way ANOVA (analysis of variance) followed by Tukey’s post-hoc test). Lowercase letters refer to comparison between control treatments, capital letters refer to comparison between 1-MCP treatments. Asterisks indicate significance between control and 1-MCP conditions referring to the same priming treatment (* *p* < 0.05, ** *p* < 0.01, Student’s *t*-test). Values represent the average ± standard deviation of five biological replicates.

**Figure 7 ijms-20-01052-f007:**
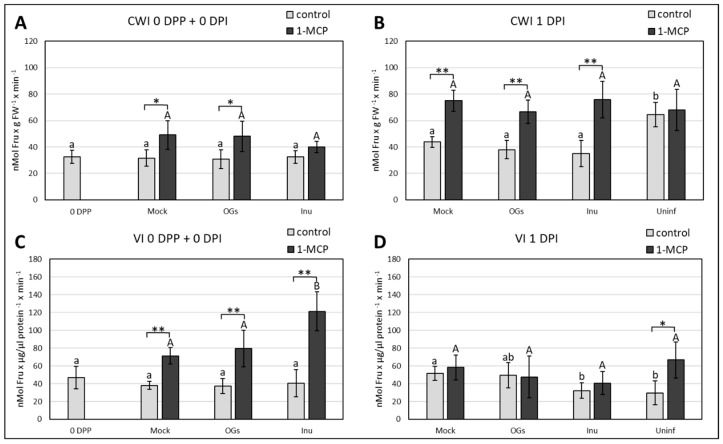
Acidic invertase activities in primed and 1-MCP treated samples. Mock = water-primed leaves; OGs = OGs-primed leaves; Inu = chicory inulin-primed leaves; Uninf = water-primed leaves inoculated with a mock solution (1/2 strength Potato dextrose broth). (**A**) CWI (Cell Wall Invertase) activities at 0 DPP and 0 DPI; (**B**) CWI activities at 1 DPI; (**C**) VI (Vacuolar Invertase) activities at 0 DPP and 0 DPI; (**D**) VI activities at 1 DPI. Bars marked with same letter(s) are not significant between each other, bars that have no letters in common indicate significance between treatments (*p* < 0.05, one-way ANOVA followed by Tukey’s post hoc test). Lowercase letters refer to comparison between control treatments, capital letters refer to comparison between 1-MCP treatments. Asterisks indicate significance between control and 1-MCP conditions referring to the same priming treatment (* *p* < 0.05, ** *p* < 0.01, Student’s *t*-test). Values represent the average ± standard deviation of five biological replicates.

**Figure 8 ijms-20-01052-f008:**
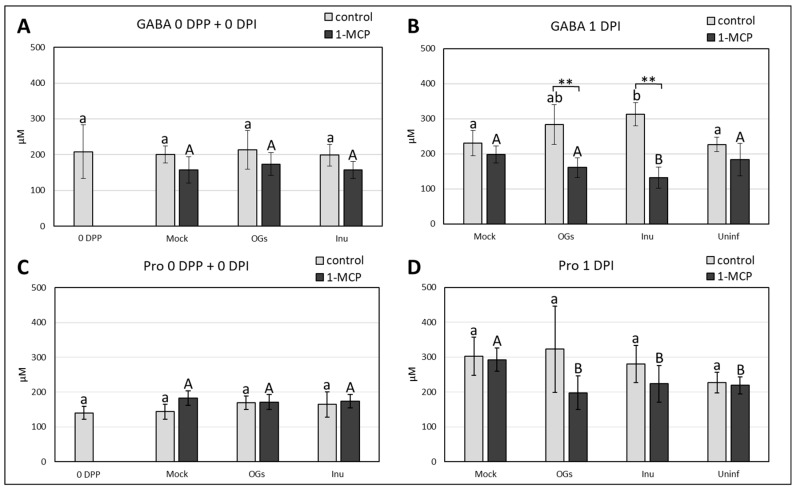
γ-Aminobutyric acid (GABA) and Pro in primed and 1-MCP treated samples. Mock = water-primed leaves; OGs = OGs-primed leaves; Inu = chicory inulin-primed leaves; Uninf = Water-primed leaves inoculated with a mock solution (1/2 strength Potato dextrose broth). (**A**) GABA levels at 0 DPP and 0 DPI; (**B**) GABA levels at 1 DPI; (**C**) Pro levels at 0 DPP and 0 DPI; (**D**) Pro levels at 1 DPI. Bars marked with same letter(s) are not significant between each other, bars that have no letters in common indicate significance between treatments (*p* < 0.05, one-way ANOVA followed by Tukey’s post-hoc test). Lowercase letters refer to comparison between control treatments, capital letters refer to comparison between 1-MCP treatments. Asterisks indicate significance between control and 1-MCP conditions referring to the same priming treatment (** *p* < 0.01, Student’s *t*-test). Values represent the average ± standard deviation of five biological replicates.

**Figure 9 ijms-20-01052-f009:**
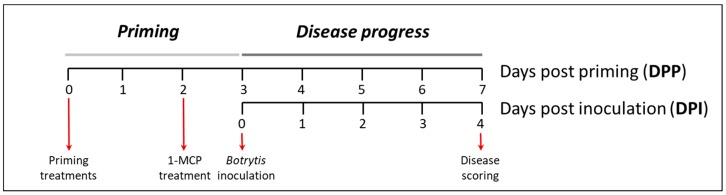
Schematic representation of the experimental design used for the priming and disease scoring performed in this study.

**Table 1 ijms-20-01052-t001:** Description of the severity classes implemented in the disease scorings done in this work with corresponding representative pictures. Scale bars =1 cm.

Class	1	2	3	4	5	6
picture	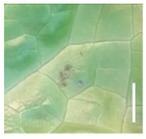	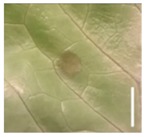	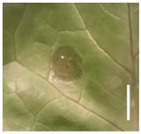	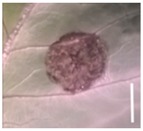	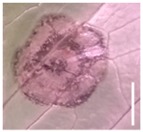	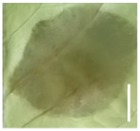
description	- Failed penetration of the pathogen. - Small and irregular signs of necrosis unable to converge in a unique infection spot.	- Spreading spot with irregular form, limited to the surrounding mesophyll. - Light brown shape.	- Spreading spot with defined circular shape, confined between the secondary veins. - Brown and moisty shape.	- Spreading lesion with circular sharp, spreading from the mesophyll to the secondary veins. - Brown and moisty shape.	- Spreading lesion with circular or elliptical shape. - Brown and moisty shape, lesion area characterized by rings of necrotic tissue.	- Spreading lesion with elliptical or irregular shape, widely spreaded across secondary veins and affecting the midrib. - Dark-brown and moisty shape with sporulation spots.
size	<0.2 cm^2^	Between 0.2 and 0.5 cm^2^	Between 0.5 and 1 cm^2^	Between 1 and 2 cm^2^	Between 2 and 4 cm^2^	>4 cm^2^

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
