# Peer review of "Sweet Immunity: Inulin Boosts Resistance of Lettuce (Lactuca sativa) against Grey Mold (Botrytis cinerea) in an Ethylene-Dependent Manner"

_ijms, 2019, doi:10.3390/ijms20051052_

Round 1
Reviewer 1 Report
The topic of the manuscript that is written by Tarkowski and co-workers is very interesting. The results of the research presented in the manuscript are valuable and should be published in the IJMS. Below there are listed points, which should be taken into account by Authors in order to improve the manuscript:
Abstract
Overall: please focus on the results of your research
Line 15, please delete “deep”, should be only “influence”
Line 15-16, please delete the sentence from the abstract and transfer to the introduction. Besides, please clearly write why fructans are typically associated with stress responses in fructan accumulating species?
Line 17-18, sentence should be rephrased better “We used commercially available inulin-type fructans for priming, a strategy that boosts plant immunity? towards subsequent pathogen attacks and/or abiotic stresses.” Below I explain what priming is?
Plants are prone to various abiotic and biotic stresses, exposure of plants to one stress affects their response during the next stress leading to enhanced defense mechanisms to a later stress. This phenomenon called as “priming” results in a faster and stronger induction of basal defense mechanisms upon subsequent biotic stress factors.
Line 28, please change in the sentence “It is the first time that the induction of pathogen resistance by …..”, it will better “It is the first time that the induction of defense mechanisms of plants on pathogen…”
Line 29-30 in my opinion it sentence should be change because this is very hypothetical “We discuss this in the context of a possible recognition of fructans as Damage or Microbe Associated Molecular Patterns” but worth mentioning in the discussion.
Line 47-48, please add abscisic acid yet. “In general, it is accepted that ethylene, JA and abscisic acid play a major role in modulating resistance against necrotrophs”
Line 49 Authors write “….in modulating resistance against ….”
Please, carefully analyze the whole manuscript point by point and consider if you can talk about resistance
Line 63-64, please insert the references “However, few research efforts focused on priming with non-structural soluble sugars.”(…)
Line 81 delete “in’, too much
Line 94-99, Sentences require changes, some should be removed because Introduction is not a Conclusion. Therefore, please clearly write the goals of the research.
Results
Overall: Sentences with references in Results section should be delete e.g. line 102-103 and other, please transfer to Discussion
Line 171, please write correctly what is Uninf? (Uninf = Mock-primed leaves not
inoculated with Botrytis, only H2O?)
Line 202, description of Figure 5 “Inhibition of ethylene signaling does not alter basal susceptibility to Botrytis, but is required for priming-induced resistance”, again “resistance”?, please do not hypothesize, just focus on the results
Line 249, please change it “…with immune processes”
Line 225 please write correctly “can be ascribed”
Discussion
Line 348-349 please change and write correctly “However, it is documented that the OGs-triggered H2O2 burst is not required for Arabidopsis resistance to Botrytis [95]”
Overall: in the whole text please write correctly
Line 474, Figure 2, please note that when appear "disease scoring", it is not the phase of infection, it is the phase of the disease
Author Response
- Reviewer 1 -
Abstract
Overall: please focus on the results of your research
Line 15, please delete “deep”, should be only “influence”
Line 15-16, please delete the sentence from the abstract and transfer to the introduction. Besides, please clearly write why fructans are typically associated with stress responses in fructan accumulating species?
Line 17-18, sentence should be rephrased better “We used commercially available inulin-type fructans for priming, a strategy that boosts plant immunity? towards subsequent pathogen attacks and/or abiotic stresses.” Below I explain what priming is?
Plants are prone to various abiotic and biotic stresses, exposure of plants to one stress affects their response during the next stress leading to enhanced defense mechanisms to a later stress. This phenomenon called as “priming” results in a faster and stronger induction of basal defense mechanisms upon subsequent biotic stress factors.
Line 28, please change in the sentence “It is the first time that the induction of pathogen resistance by …..”, it will better “It is the first time that the induction of defense mechanisms of plants on pathogen…”
Line 29-30 in my opinion it sentence should be change because this is very hypothetical “We discuss this in the context of a possible recognition of fructans as Damage or Microbe Associated Molecular Patterns” but worth mentioning in the discussion.
- Thank you. We made modifications to the abstract according to your suggestions and we deleted the final sentence as requested.
Introduction
Line 47-48, please add abscisic acid yet. “In general, it is accepted that ethylene, JA and abscisic acid play a major role in modulating resistance against necrotrophs”
- Thank you, changes have been done accordingly.
Line 49 Authors write “….in modulating resistance against ….”
- Thank you, we replaced the word resistance by carefully considering where it was not pertinent.
Please, carefully analyze the whole manuscript point by point and consider if you can talk about resistance
- Thank you, we went throughout the text and weakened our statements, as requested.
Line 63-64, please insert the references “However, few research efforts focused on priming with non-structural soluble sugars.”(…)
- Thank you, we inserted appropriate references.
Line 81 delete “in’, too much
- Thank you, this was changed accordingly.
Line 94-99, Sentences require changes, some should be removed because Introduction is not a Conclusion. Therefore, please clearly write the goals of the research.
- Thank you, we changed this part of the text and explicitly stated the research goals.
Results
Overall: Sentences with references in Results section should be delete e.g. line 102-103 and other, please transfer to Discussion
- Thank you, sentences with references have been deleted.
Line 171, please write correctly what is Uninf? (Uninf = Mock-primed leaves not
inoculated with Botrytis, only H2O?)
- Thank you, we have clarified it now. We indeed referred to water primed leaves, subjected to a mock inoculation (consisting in half strength potato dextrose broth).
Line 202, description of Figure 5 “Inhibition of ethylene signaling does not alter basal susceptibility to Botrytis, but is required for priming-induced resistance”, again “resistance”?, please do not hypothesize, just focus on the results
- Thank you, we re-phrased this description.
Line 249, please change it “…with immune processes”
- Thank you, this was changed accordingly
Line 225 please write correctly “can be ascribed”
- Thank you, changes have been done accordingly
Discussion
Line 348-349 please change and write correctly “However, it is documented that the OGs-triggered H2O2 burst is not required for Arabidopsis resistance to Botrytis [95]”
- Thank you, changes have been done accordingly
Overall: in the whole text please write correctly
- Thank you, we carefully went through the text in order to improve the English quality.
Line 474, Figure 2, please note that when appear "disease scoring", it is not the phase of infection, it is the phase of the disease
- Thank you, we attached a new version of the figure where we amended this mistake. We changed “days post infection” by “days post inoculation” throughout the manuscript

Reviewer 2 Report
The manuscript by Tarkowski et al. describes the priming effect of OGs and inulin type fructans in lettuce and demonstrates that treatments with these substances reduce disease symptoms of gray mold in the pathosystem Lactuca sativa-Botrytis cinerea. Furthermore, they demonstrate the involvement of ethylene signalling in this priming response. In addtion to ethylene, a number of factors known to play a role in the activation of plant defense responses against necrotrophs are also considered and evaluated in the pathosystem investigated.
I find the work being presented solid and interesting.The rationale behind it is strong, the experimental design is correct, the methods are sound and an interesting body of information has been generated. The results obtained have been elaborated and discussed seriously and I think the manuscript constitutes a useful addition to literature, offering relevant information about the description and characterization of the priming triggered by a group of natural compounds in a particular pathosystem, information which is appropriately integrated and disussed in the context of the analysis of the plant defense responses and might be useful for the scientific community working in the field. It widens our basic knowledge of priming and plant defense mechanisms and also provides pieces of information on which possibilities of control can be developed.
The manuscript is well written and prepared and to my opinion it deserves consideration for publication in International Journal of Molecular Sciences. My recommedation is “Accept after minor revision”. The manuscript is nicely and coherently organized and scientifically I find it serious and rigurous. I think it essentially needs a detailed review for text editing (i.e. line 81) and correct calling for figures in the text (i.e. line 217: it is indicated “Figure 7”, but … should not be “Figure 6”?).
I major formal problem concerns the list of references. In my first reading I found that some references were not being correctly used and cited in the text. Once revised in detail, I think the reason for it is that there is a problem with the numbering. From reference 11 onwards the numbers given in the text to call for references in the list of references do not correspond to the right references. I think it is a problem generated when handling the references list and the numbers asigned to each reference by the bibliography editor. It certainly has to be solved. In addition to this problem I find that two references in the list, 14 and 15 following the numbering given in the text, are never cited (¿…?). Authors should carefully check these problems to make the reference list coherent.
Author Response
- Reviewer 2 -
The manuscript is well written and prepared and to my opinion it deserves consideration for publication in International Journal of Molecular Sciences. My recommedation is “Accept after minor revision”. The manuscript is nicely and coherently organized and scientifically I find it serious and rigurous. I think it essentially needs a detailed review for text editing (i.e. line 81) and correct calling for figures in the text (i.e. line 217: it is indicated “Figure 7”, but … should not be “Figure 6”?).
- Thank you, thanks also to reviewer 1 we went through the text in order to improve the English quality and correction of grammar mistakes. We also fixed the numbering of the figures in the text.
I major formal problem concerns the list of references. In my first reading I found that some references were not being correctly used and cited in the text. Once revised in detail, I think the reason for it is that there is a problem with the numbering. From reference 11 onwards the numbers given in the text to call for references in the list of references do not correspond to the right references. I think it is a problem generated when handling the references list and the numbers asigned to each reference by the bibliography editor. It certainly has to be solved. In addition to this problem I find that two references in the list, 14 and 15 following the numbering given in the text, are never cited (¿…?). Authors should carefully check these problems to make the reference list coherent.
- Thank you. Indeed, we had a problem with the reference editor. We went carefully through all the reference list and fixed it.

Round 2
Reviewer 1 Report
The manuscript was improved by the Authors. I would like to thank the Authors for taking into account all my suggestions. I have a few comments yet
Abstract
Line 30-31, Please, at the end of the abstract, insert the sentence again, I quote "We discuss our results in the context of a possible recognition of fructans as Damage or Microbe Associated Molecular Patterns”
Introduction
Line 48, please add symbol ET after ethylene, i.e. “(ABA) and ethylene (ET) in a pathosystem-specific manner. In general, it is accepted that ET”
Line 96, please delete one space “of inulin”
Results
Line 100, Botrytis should be italic
Overall: During the description of the results, an impersonal form should be used
For example: Line 100-104, line 244 etc.
Line 153 -155, these sentences should be transfer to discussion and the sentence can be started “Histochemical measurements of H2O2 accumulation in primed and infected leaves using the DAB were performed.
Author Response
- Reply to reviewer 1 -
Abstract
Line 30-31, Please, at the end of the abstract, insert the sentence again, I quote "We discuss our results in the context of a possible recognition of fructans as Damage or Microbe Associated Molecular Patterns”
Thank you, we inserted the sentence again.
Introduction
Line 48, please add symbol ET after ethylene, i.e. “(ABA) and ethylene (ET) in a pathosystem-specific manner. In general, it is accepted that ET”
Thank you. However, between ethylene experts this abbreviation is not very well accepted, as it is not an acronym as compared to other hormones names (ABA= ABscissic Acid; SA= Salycilic Acid; JA= Jasmonic Acid). After consulting, we would like to ask if we can keep the extended form for ethylene for consistency with the guidelines of the experts in the ethylene field.
Line 96, please delete one space “of inulin”
Thank you, we amended to this typo.
Results
Line 100, Botrytis should be italic
Thank you, changes were done accordingly.
Overall: During the description of the results, an impersonal form should be used
For example: Line 100-104, line 244 etc.
Thank you, we went through the results and fixed this accordingly.
Line 153 -155, these sentences should be transfer to discussion and the sentence can be started “Histochemical measurements of H2O2 accumulation in primed and infected leaves using the DAB were performed.
Thank you, changes were done accordingly, we also corrected references numeration following the shift of the sentence.
Finally, we would like to thank Reviewer 1 for his/her valid inputs, that contributed to significantly ameliorate the quality of our manuscript.
